# A Slow-Release Fertilizer of Urea Prepared via Melt Blending with Degradable Poly(lactic acid): Formulation and Release Mechanisms

**DOI:** 10.3390/polym13111856

**Published:** 2021-06-03

**Authors:** Mujtahid Kaavessina, Sperisa Distantina, Esa Nur Shohih

**Affiliations:** Chemical Engineering, Universitas Sebelas Maret, Surakarta 57126, Indonesia; sperisa_distantina@staff.uns.ac.id (S.D.); esanurshohih@gmail.com (E.N.S.)

**Keywords:** poly(lactic acid), urea, melt blending, slow-release fertilizer

## Abstract

In this research, a low molecular weight poly(lactic acid) (or PLA) synthesized from direct polycondensation was melt compounded with urea to formulate slow-release fertilizer (SRF). We studied the influence of the molecular weight (*M*_W_) of PLA as a matrix and the urea composition of SRF towards release kinetics in water at 30 °C. The physical appearance of solid samples, the change in urea concentration, and acidity (pH) of water were monitored periodically during the release test. Three studied empirical models exhibited that diffusion within the matrix dominated the urea release process, especially when the release level was less than 60%. Thus, a lower *M*_W_ of PLA and a higher urea content of SRF showed a faster release rate. For the entire length of the release experiment, a combination of diffusion and degradation mechanisms exhibited the best agreement with the experimental data. The hydrolytic degradation of PLA may begin after 96 h of immersion (around 60% release level), followed by the appearance of some micro-holes and cracks on the surface of the SRF samples. Generally, this research revealed the good release performance of urea without residues that damage the soil structure and nutrient balance.

## 1. Introduction

The global consumption of agricultural products has steadily increased proportionally with world population growth. Rice, maize, and wheat are the most important cereals worldwide in terms of production. Nowadays, agricultural intensification is the main alternative that encourages farmers to increase agricultural production with limited agricultural land. Exploiting natural resources, such as soil, water, space, or energy, is necessary for every stage of large-scale agriculture. Many reports have described the depletion of organic matter, chemical contamination of soil, decreased soil fertility, and water spring deterioration related to agricultural products [1,2,3]. The main challenge has become to increase the quantity and quality of crops product via sustainable agriculture.

Fertilization is an effort to restore soil fertility that plays an important role in crop production. Thus, it contributes primarily and directly to the production costs. Pypers et al. reported that the key to successful plant fertilization is the appropriate dosage and timing of fertilization [4]. Improper fertilization techniques, inappropriate fertilization times, and both excessive and insufficient fertilizer doses contribute to detrimental effects on the environment. Indeed, this condition affects the quality and quantity of agricultural products.

Urea is very widely used in agriculture, known as nitrogen fertilizer, because of its high nitrogen content (46%). Nitrogen is a necessary nutrient for plant growth, and it is the most crucial factor commonly considered to be yield-limiting. The conversion mechanism of how urea becomes nitrogen absorbable by plants in the form of ammonium (NH_4_^+^) and nitrate (NO_3_^−^) is known well [5,6]. The urease enzyme in moist soil will encourage the nitrogen in urea to be converted into ammonium (NH_4_^+^) via hydrolysis. In the nitrification process, ammonium is converted into nitrite (NO_2_^-^) and then to nitrate (NO_3_^-^) by oxidation [5]. However, many factors can easily eliminate both substances (NH_4_^+^ and NO_3_^-^) from soils, such as drainage; denitrification of nitrate-producing nitrous oxide gas (N_2_O), nitric oxide gas (NO), or nitrogen gas (N_2_); nitrogen volatilization; and surface run-off [7]. Thus, it has been estimated that only 30–50% of the nitrogen in urea can be absorbed by plants [8,9].

Many efforts have been studied and applied to reduce the loss of nitrogen and to conserve and protect our environment, such as (i) fertilization management: integrated and site-specific management, and balanced fertilization: (ii) chemical additives such as nitrification inhibitors; and (iii) modification of fertilizer properties: controlled/slow-release fertilizer (CRF or SRF) [10,11]. In the last decade, CRF/SRF has become an exciting topic for researchers in academia and industry.

SRF is the type of fertilizer that releases nutrient elements slowly and regularly, approaching the absorption patterns of plants. The nutrient elements contained in the fertilizers do not get carried away by the water. The synthesis of SRF combines fertilizer (such as urea) and other materials with water retention properties. Recently, three methods were developed to produce SRF, i.e., (i) chemically combined fertilizers, (ii) coated fertilizers, and (iii) physically blended fertilizers [8,9,12].

In SRF formulation, commercial or developed SRFs mostly utilize materials such as urea–formaldehyde (UF), sulfur, zeolite or modified zeolite, bentonite, polyolefin, polyvinylidene chloride, polystyrene, etc. These materials are used alone or in combination with others as coatings, matrices, carriers, or grafted materials in SRFs [12,13,14,15,16], which do not easily degrade properly in the soil. These accumulated residues of SRFs allow damage to the soil structure and nutrient balance in the soil. Therefore, the research focus trend has been switched to exploiting safer and environmentally friendly materials that can also control the release rate of SRF.

This problem inspired the idea of utilizing low molecular weight poly(lactic acid) as a fertilizer carrier matrix. As known, poly(lactic acid) is not polluting to the environment after it has naturally degraded in a humid environment or a solution. It could decompose into natural products/biomass and gasses that are not harmful/toxic to the crop plants [17,18]. Thus, there is no residual accumulation in the use of this material in SRF formulations.

In our previous work, the degradation rate of poly(lactic acid) or PLA was affected by other polymers or substances in blends or its molecular weight [19,20]. Based on the results, we studied the possibility of developing fertilizer by utilizing low molecular weight PLA as a substitute for the existing matrices of SRF. The objectives of this research were: (i) to formulate slow-release fertilizer (SRF) of urea by exploiting the potential properties of low molecular weight (*M*_w_) PLA as a matrix, and (ii) to study the urea release mechanisms of SRF through three mathematical model approaches.

We blended micro-size urea into the melt of low molecular weight PLA obtained through direct polycondensation of lactic acid to achieve the objectives. Different loadings of urea in matrix and molecular weights of PLA were analyzed regarding their release behavior. The presence of urea in the SRF was detected by Fourier transform infra-red (FTIR). The release of urea in the SRF was studied through a static release experiment designed mainly according to the other research methods [8,9]. The concentration of urea in the solution was recorded, as well as its acidity (pH). Before and after the release test, a morphological analysis of the samples was conducted by scanning electron microscopy (SEM).

## 2. Materials and Methods

The lactic acid in a 88–90% aqueous solution was produced by Scharlau (Barcelona, Spain) with a density of 1.20 (20°/4°). Stannous (II) chloride dihydrate (98%), urea powder, and chloroform were ordered from Sigma-Aldrich (Jakarta, Indonesia). Methanol was produced by Avonchem (Macclesfield, UK). All chemicals were used as received without any additional purification.

Direct polycondensation of lactic acid was carried out without any solvents in the 500 mL flat–bottom 3–necked flasks completed by a Dean–Stark trap. Nitrogen flowed into this flask through a capillary inlet. The reaction condition was controlled at 138 °C and stirred at 150 rpm using a magnetic heat stirrer, RCT Basic IKAMAG^®^ safety control. Stannous (II) chloride as the catalyst was added at about 0.1 wt%.

Micro–sized urea was blended in a micro–compounder at 50 rpm and 110 °C for 1 min. The granulation process was carried out by dripping the molten SRF on a tray. The nomenclature of samples prepared and analyzed in this investigation is shown in Table 1.

The average molecular weight of synthesized PLA was determined at 30 °C by a Waters Alliance GPCV 2000 system. Tetrahydrofuran (THF) as the mobile phase was set at a flow rate of 1 mL/min. The presence of urea in formed SRF was detected using a Perkin-Elmer 630 IR spectrophotometer (FTIR) within the IR spectrum range of 4000–400 cm^–1^.

A static release experiment was performed at room temperature (around 30 °C). Figure 1 depicts the experimental apparatus for determining the static release of urea in water, emulating previous research [8,9]. A small magnetic stirrer bar (3 mm diameter and 6 mm long) was used to stir the samples at 50 rpm. SRF samples (3 g) were put into a tube, 25 mm long and 5 mm in diameter, with one end closed. The tube containing the SRF was placed horizontally in a glass beaker (150 mL) filled with 100 mL of water. Periodically, the urea concentration and the acidity (pH) of water were recorded. Urea was detected using a Genesis 20 Visible spectrophotometer (Thermo Scientific, Waltham, MA, USA) operating at a wavelength of 440 nm assisted by Ehrlich reagent. The urea concentration was calculated using a standard curve that correlated the urea concentration and absorbency value on the spectrophotometer reading. The degraded solids of SRFs were observed regarding their morphology via scanning electron microscopy (SEM), JEOL JSM-6360A (Tokyo, Japan), at 15 kV.

Three mathematical models were applied to analyze the release mechanism by fitting the curve of the fractional release, i.e., (i) the Korsmeyer–Peppas model, (ii) the diffusion–relaxation model, and (iii) the diffusion–degradation model. OriginPro software 2016 assisted in plotting the nonlinear fit of the three models to determine the parameter constants.

The first model only considers the diffusion that occurred during urea release, as presented below [21,22]:(1)$$\frac{M_{t}}{M_{\infty }}=k\;t^{n}\;$$
where *M_t_* is the amount of urea released at time *t* (g), *M*_∞_ is the amount of urea released over an infinite time or the total amount of urea when it is all released from the SRF sample (g), and *t* is the time of urea release (m). The *k* value is the kinetic constant, combining the characteristics of the urea–SRF system, and *n* is the release exponent, representing a transport mechanism, whereas *M_t_*/*M*_∞_ refers to the fraction of urea released in water at time *t*. In the diffusion–relaxation model, 2 constants refer to the diffusion and the relaxation, as formulated below [8,23,24]:(2)$$\frac{M_{t}}{M_{\infty }}=k_{1}t^{m}+k_{2}t^{2m}\;$$
where *k*_1_ and *k*_2_ are associated with diffusion and relaxation, respectively. The *m* value is determined to be 0.43, based on the geometric shape of SRF representing the diffusion exponent [23]. For the diffusion–degradation model, there is 1 constant related to diffusion and 3 constants related to degradation, as defined below:(3)$$\frac{M_{t}}{M_{\infty }}=at^{0.5}+bt+ct^{2}+dt^{3}\;$$
where *a* is associated with diffusion and the 3 constants (*b*, *c* and *d*) are associated with degradation.

## 3. Results and Discussion

Lactic acid was polymerized solely without any solvents through direct polycondensation. Stannous chloride dihydrate (SnCl_2_.2H_2_O) was added as the catalyst and the temperature was set at 138 °C during polymerization. As seen in Table 1, the average molecular weight of poly(lactic acid) obtained varied in accordance with the polymerization time. The polycondensation time of lactic acid varied at 16, 24, and 32 h and resulted in an average molecular weight of 6015.2 Da, 10,264.7 Da, and 13,564.2 Da, respectively. Further, this obtained poly(lactic acid) was blended with micro-sized urea to make slow-release fertilizer (SRF), as summarized in Table 1.

### 3.1. Molecular Structure of Slow-Release Fertilizer (SRF)

The SRF’s structure was investigated using an infra-red (IR) spectrophotometer to verify urea and PLA’s successful blending through melt blending. Figure 2 shows the IR spectra of some samples. The neat PLA sample (Figure 2A) was also scanned to determine urea’s presence in slow–release fertilizer. Five dominant peaks show the functional group of poly(lactic acid). The wavenumber around 870 cm^−1^ shows the peak representing the bond of −C−C−. This peak also indicates the semi-crystalline phase of the obtained PLA. The methyl groups −CH− or −CH_3_ appear at the wavenumber around 2944 cm^−1^ and 1382 cm^−1^ with different vibration modes. Garlotta [25] explained that stretching and bending modes are represented by the peaks at 2944 cm^−1^ and 1382 cm^−1^, respectively. The peaks at the wavenumbers around 1740 cm^−1^, 1093 cm^−1^ and 1182 cm^−1^ represent the carboxyl group’s presence, i.e., C=O and C−O with the same vibration mode (stretching).

Only three peaks appear on the IR spectra of the SRF samples, i.e., around 3472 cm^−1^, 1585 cm^−1^, and 1560 cm^−1^ (Figure 2B, C and D). These peaks represent the groups of N−H stretching, N−H deformation, and C−N stretching, respectively [26]. Based on this analysis, the urea in slow-release fertilizer can be detected and proven qualitatively.

### 3.2. Urea Release Behavior

Further, all samples were then tested in the static release apparatus (Figure 1) to study their urea release behavior in water. This test provides two data simultaneously relating to the change in the urea concentration and the acidity (pH) of water. Appendix A tabulates data on the urea concentration in water during the release test. These data were then processed to calculate the accumulated fraction of urea released in water during the immersion, as depicted in Figure 3. The release fraction presents information on the amount of urea released at time t compared with the total urea in the SRF sample. As observed during the release test, there is no visible swelling of SRF.

Figure 3 can be divided virtually into two zones (A and B). As seen, urea’s release appears to find the release equilibrium at around 75%. For all samples, the slope of urea release in Zone A is sharper than that in Zone B. This shows that urea was released rapidly in the first stage (Zone A), then the release rate tended to be slow in Zone B. For example, the fraction of released urea in Zone A for SRF101 changed by about 25% within 120 h (in the range of 48–168 h). In Zone B, it required a time of around 168 h (in the range of 168–336 h) to achieve an additional urea release of 12.5%. The increasing concentration of urea in SRF urged the urea release to be faster. After immersion for 96 h, the percentage of urea release reached about 59%, 66%, and 72% for urea concentrations of 1% (SRF201), 3% (SRF203), and 5% (SRF205), respectively. This phenomenon proves that urea, with its hygroscopic property, still existed and affected the release process.

The other phenomenon that can be highlighted is the molecular weight (*M*_W_) of poly(lactic acid) itself. This parameter describes the length of PLA chains, which have different properties. Utilizing the higher *M*_W_ of poly(lactic acid) tended to inhibit the urea release. After immersion for 96 h, the percentage of urea release was monitored at 63%, 59%, and 57% when the Mw of PLA was 6015.2 Da (SRF101), 10,264.7 Da (SRF201), and 13,564.2 Da (SRF301), respectively. This showed that the permeability of PLA decreased with increasing molecular weight so that the contact of water and urea in the PLA matrix was increasingly inhibited. Further explanations are discussed in the modeling section.

The utilization of the low molecular weight poly(lactic acid) as a matrix of SRF aimed to exploit its degradable property. Qi et al. reported a review of the biochemical processes of PLA degradation. They concluded that those processes mainly included chemical hydrolysis and biodegradation in the natural soil microcosm [17]. The presence of ester bonds in PLA can be broken with the chemical hydrolysis that may occur during the PLA’s immersion. Carboxylic acid and alcohol arise as a result of breaking the ester bonds. Indeed, the existence of carboxylic acid influences the acidity of the solution. Instead, the urea initially tends to be alkaline when it dissolves in water [27]. Thus, the monitored pH values of solutions describe the result of combining properties between carboxylic acid and urea dissolved in water. The changes in the solution acidity are tabulated periodically in Table 2.

In Table 2, for neat PLA, the acidity (pH) tends to be constant or decrease slightly in the time range between 0 and 96 h, then becomes relatively more apparent with increasing time above 96 h. This means that the hydrolytic degradation may begin after 96 h, which is indicated by the release of acid resulting from scission of the PLA chain. All samples of SRF showed the same tendency. The pH increased gradually and was followed by a decrease during the range of immersion time. This exciting phenomenon could be explained by the urea release causing the increasing pH of the solution, then the acid from PLA degradation decreasing the pH solution. This statement will be analyzed using the mathematical models, as discussed in this article.

By using pH values, the initial degradation time can be observed at different times. The degradation time of SRF101, SRF201 and SRF301 was initiated around 60, 72 and 72 h, respectively. This means that increasing the molecular weight of PLA caused a shift to the longer initial degradation time. This statement is confirmed by the morphological sample after immersion at a specific time.

### 3.3. Modeling of Urea Release Behavior

Some researchers have reported several mathematical models associated with the release mechanisms of an active substance from a matrix. These models were developed via different approaches, considering (i) only the diffusion and (ii) the combination of diffusion and other factors such as relaxation and erosion/degradation [22,23,28]. In this article, three mathematical models were used to analyze the release mechanism by fitting the curve of the fractional release, i.e., the Korsmeyer–Peppas model, the diffusion–relaxation model, and the diffusion–degradation model. We examined and verified the fit of the curves of the experimental data with these developed models. The proper model will be applied to describe the release mechanism and explain the studied variables, i.e., the molecular weight of poly(lactic acid) and the urea concentration in SRF.

The Korsmeyer–Peppas model is a simple exponential expression to analyze the controlled release behavior of an active substance from its matrices. Table 3 recapitulates the data from fitting the curve of the fractional release of urea using the Korsmeyer–Peppas model. This model elaborates the values of *n* depending on the geometric shape of the sample. For the spherical form, *n* < 0.43 corresponds to Fickian diffusion, while 0.43 < *n* < 0.85 represents anomalous transport (non-Fickian diffusion) [21]. It can be seen that all samples of SRF exhibit Fickian diffusion. This table also presents *R^2^*, which shows how close the data are to the fitted regression line. Based on the *R^2^* values, all the samples’ release curves have good enough agreement with this model.

Peppas et al. already explained that this equation is accurate for the first 60% of a release fraction curve [21]. This explanation agrees with our results, as shown in Figure 4 and Appendix A, which depicts the urea fraction released versus time. For more than 60%, the difference in the data between the experimental results and the model calculation is relatively large. It indicates that diffusion transport dominates in the first 60% of release for all SRF samples. Referred to as the Fickian diffusional release, this mass transfer occurs by the usual molecular diffusion of urea due to the gradient of chemical potential.

The first model describes only the initial kinetic behavior (the release level is less than 60%). We have already analyzed the matrix’s morphology solely during the immersion to explain the release behavior over the entire range of immersion time (0–504 h). Figure 5 shows the SEM images of neat PLA before (Figure 5A) and after immersion in water for 168 h (Figure 5B) and 504 h (Figure 5C). The SEM image in Figure 5A shows a difference in polymer density, indicating the crystalline and amorphous phases in solid poly(lactic acid). The presence of the amorphous phase looks whiter in color and has cracks (shown by arrows). SEM images of PLA after the degradation test in water show significant changes, as seen in Figure 5B,C. The PLA surface became rough and developed numerous micro-holes along the length of the degradation time.

In a previous study, some researchers reported that water diffusion into the amorphous phase initiates the PLA’s hydrolytic degradation in aqueous or humid environments. This process involves the scission of PLA chains that are dominant in ester bonds concentrated in this phase to generate a lower *M*_W_ PLA or monomer (lactic acid). Thus, the degradation occurs preferentially in the amorphous phase and then continues to the crystalline phase [29,30,31]. The rough surface and numerous micro-holes could be ascribed to PLA chain scission and removal in both phases.

In the above explanation, PLA as a matrix is degraded during the hydrolytic degradation test after a certain period. This phenomenon is strongly suspected of affecting urea’s release from the matrix, especially after immersion above 96 h when the solution’s pH tends to decrease more significantly (see Table 2 for neat PLA). Figure 4 shows that the release level of urea is about 60% after immersion for 96 h. When correlated with the Korsmeyer–Peppas model, several other factors influenced release after 96 h. Thus, we carried out a morphological analysis of SRF samples after immersion for a specific time (Figure 6). This analysis is expected to support a mathematical model that depicts the urea’s release throughout the range of time studied.

Figure 6A shows the SRF201’s morphology before the process of urea release. The distribution of micro-sized urea is evenly distributed in the PLA matrix with little aggregation being formed. This indicates that the stirring process can disperse the urea. After immersion for 96 h (see Figure 6B), several holes appeared to be forming, showing a degradation of the polymer matrix. The holes became enlarged and the degradation effect became more visible with increasing immersion times of 168 h (Figure 6C) and 504 h (Figure 6D). The existence of these holes may be caused by (i) the initial degradation of PLA in the amorphous area or (ii) the release of urea aggregate (if any) in the SRF samples.

Figure 6 shows that the morphological changes in SRF became more significant above 96 h, but the released urea fraction tended to be less (see Figure 3). This phenomenon illustrates the possibility of different urea release mechanisms before PLA degradation and when the PLA degradation occurred.

As mentioned above, we also examined and verified the curve fit of the experimental data with two models, i.e., the diffusion–relaxation model and the diffusion–degradation model. Both models are used to further describe the release behavior over the entire period of the release time.

Figure 4 depicts the urea release as a function of time for SRF101 based on experimental data and the calculation data. It can be seen that the empirical model also has better agreement with experimental release data when relaxation or degradation are considered as parameters in the model. All constants related to both the relaxation and degradation parameters are analyzed and tabulated in Table 4 and Table 5. Both tables also tabulate the calculated data’s R-squared (*R*^2^), known as the coefficient of determination.

Table 4 presents the obtained parameters from the data analysis using the diffusion–relaxation model. It can be seen that there is a large gap between the *k*_1_ and *k*_2_ constants. Besides that, the *k*_1_ value is always higher than *k*_2_. This fact indicates that the diffusion of urea from the PLA matrix dominates its release. The relaxation term only has a minimal effect on diffusion. The negative sign in the *k*_2_ value indicates a correction for the dominance of diffusion in the model. The *R*^2^ value of this model shows a better-fitting curve compared with the Korsmeyer–Peppas model.

The *k*_1_ value tends to decrease proportionally to the increase in the PLA’s molecular weight (see SRF101, SRF201, and SRF301). This result verifies the previous statement quantitatively, namely that utilizing the higher *M*_W_ poly(lactic acid) tended to inhibit the urea release. The *k*_1_ value decreased from about 0.1119 to 0.1015, and 0.0981, when the PLA’s molecular weight increased from 6015.2 to 10,264.7, and 13,564.2 Da, respectively.

The *k*_1_ value also describes urea concentration’s effect on its release (see SRF201, SRF203, and SRF205). Quantitatively, the *k*_1_ value increased from about 0.1015 to 0.1284 and 0.1352 when urea concentration was increased from 0.01 to 0.03 and 0.05 g/3 g of SRF. Again, this result confirms our previous statement that urea’s hygroscopic property in the matrix is still in existence and affects its release during immersion.

Table 5 presents the constants obtained from the data analysis using the diffusion–degradation model. This model has better accuracy than the previous two models as shown by the *R*^2^ value (closer to 1). In the data analysis, the constant of *a*, which indicates the diffusion factor, has a much greater value than the other three constants (*b*, *c*, and *d*). Again, the obtained data show that diffusion was a dominant factor during urea release. Even though the constant values of *b*, *c*, and *d* are very small, they illustrate that other factors influenced the urea release, especially at the release level above 60% (see Figure 4). Because these three factors are related to degradation, it can be highlighted that poly(lactic acid) degradation also influences urea release. The effects of degradation may not be as significant as those caused by diffusion. It might be that PLA begins to degrade after 96 h of immersion, as previously described. Moreover, most of the urea in the PLA matrix had been released in the solution.

### 3.4. Urea Release Duration

Table 6 tabulates some materials explored to examine the influence on the urea release rate. These materials were utilized as encapsulating matrices, coating materials, blending materials, etc. It can be seen that different combinations of these materials gave many possibilities for the formulation of slow-release fertilizers. The addition of modifiers that act as binders, fillers, or emulsifiers had different effects on the urea release performance, depending on the property of the modifier itself or the interaction between the modifier and the primary material in the SRF. For example, the hydrophilicity of the modifier in bentonite-based SRF, hydroxypropyl methyl-cellulose (HPMC), was more hydrophilic and induced a faster urea release than that of starch [9]. A different result was reported by Pereira et al. [6], namely that the hydrophilicity causes a good interaction between polyacrylamide hydrogels and bentonite, resulting in a slower release of urea compared with polycaprolactone.

The addition of emulsifiers, such as span-80, increased the dispersity of the sealant in sulfur-based SRF. Yu and Li reported that brittle paraffin’s coating efficiency as a hydrophobic sealant was improved due to the span-80 enhancing its adhesion [15]. Both synthetic and natural polymers were also explored as coating materials or matrices. The formulation of the hydrophilicity and hydrophobicity of the polymers and modifiers significantly influenced the release pattern [16,32,33,34,35,36].

Table 6 provides an overview of several successful attempts to slow urea’s release with varying release durations. The utilization of inorganic materials and synthetic polymers in SRF raises problems on the other side. Fertilization with sulfur-coated urea (SCU) has the potential to improve soil acidity. However, polymer and minerals in SRF will leave the residue, contributing to other forms of pollution, and they are difficult to degrade properly in the soil [12]. In this study, the release duration of the obtained SRFs was about 168 h to achieve 75% urea release when tested in water. This result is comparable with the other results, as shown in Table 6. This SRF utilized the low molecular weight poly(lactic acid) without any other additives or modifiers. Thus, this fertilizer is promising because it does not leave residues that damage the soil structure and the nutrient balance in the soil. PLA can be naturally degraded into substances that are not toxic and harmful to plants [17,18].

**Table 6 polymers-13-01856-t006:** The urea release duration of slow-release fertilizer (SRF) conducted in this work and some other reports.

Material + Modifier (Additive)	Preparation Method	Release Test	* Release Duration	Ref.
Mineral Natural bentonite + binder: corn starch or hydroxypropyl methyl-cellulose Montmorillonite clay (bentonite) + hydrophobic/hydrophilic polymer: polycaprolactone or polyacrylamide hydrogel	Melt blending Melt blending	Higuchi procedure in water at 30 °C Immersed in an aqueous medium at room temperature	118 h or 48 h 30 h or 60 h	[9] [6]
Sulphur-based Phosphogypsum + paraffin wax + span-80 (as emulsifier)	Coating	Static release test in water at 25 °C	240 h	[15]
Synthetic polymer Polyurethane + mesoporous silica Polystyrene + wax Polystyrene + polyurethane	Coating Coating	Immersed in deionized water at 25 °C Immersed in deionized water at 25 °C	10–50 d 42 d 70 d	[33] [34]
Degradable synthetic polymer Polyesters: poly(hexamethylene succinate)/PHS Polyvinyl alcohol + biochar	Melt blending Melt blending	Immersed in deionized water at 25 °C Buried in soil column experiment at 25 °C	400 h 25 d	[24] [16]
Natural polymer Starch + glycerol Alginate + Κ-carrageenan/celite superabsorbent Chitosan salicylaldehyde Poly(lactic acid) with a low molecular weight	Coating Coating Solvent casting Melt blending	Buried in compost soil at 25 °C Buried in soil at 25 °C Immersed in distilled water at 25 °C Static release test in water at 30 °C	15–30 d 6 d 200 h 168 h	[32] [35] [36] This study

* Time required to achieve 75% release.

## 4. Conclusions

Slow-release fertilizer (SRF) with urea was successfully synthesized through melt blending between low molecular weight poly(lactic acid) and urea. Through the FTIR spectra and SEM images, we can confirm the presence of urea and its distribution in the SRF. To investigate the urea release mechanism of SRFs in water, we obtained the fractional release data of urea from static release experiments and we evaluated these data by fitting the curve of the fractional release through three mathematical models. It was found that a higher urea concentration in the SRF exhibited a faster release of urea. The hygroscopic property of urea could still exist and influence the release process. Utilizing the higher molecular weight poly(lactic acid) had a slower urea release due to the decreasing permeability of PLA. The low permeability inhibited the contact between water and urea in the PLA matrix. The diffusion–degradation model showed the best match between all samples’ release behavior and the mathematical approaches compared with the other two models. However, the three studied models showed the same tendency that diffusion within the matrix dominated the urea release process, especially when the release level was less than 60%. The erosion (in this case, as hydrolytic degradation) of the PLA matrix may begin after 120 h of immersion. This immersion time indicates that the urea release level was around 60%. Thus, above this level, the degradation factor began to appear and, in the model, it had the best match with the experimental data. This SRF is promising because it does not leave residues that damage the soil structure and the nutrient balance in the soil.

## Figures and Tables

**Figure 1 polymers-13-01856-f001:**
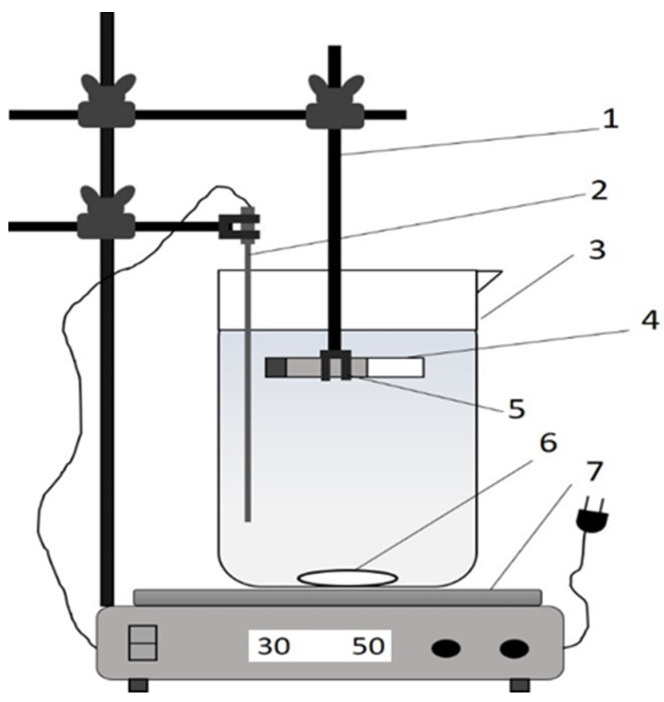
Scheme of the static release apparatus: 1 Tripod and clamp, 2 thermocouple (PT1000), 3 glass beaker, 4 sample tube, 5 SRF sample, 6 magnetic stirrer bar, and 7 magnetic heat stirrer.

**Figure 2 polymers-13-01856-f002:**
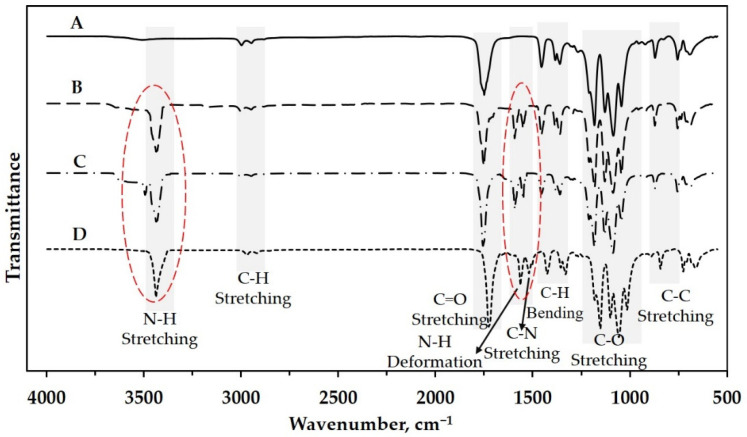
IR spectra of (**A**) neat PLA and some slow-release fertilizer: (**B**) SRF101, (**C**) SRF201, and (**D**) SRF301.

**Figure 3 polymers-13-01856-f003:**
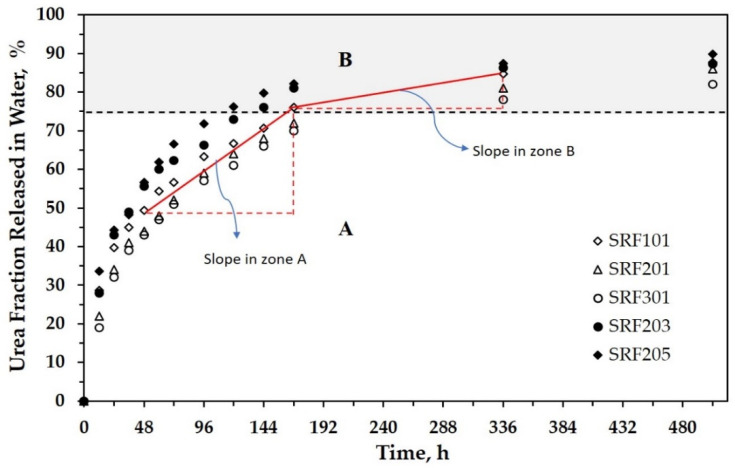
Fractional release of urea as a function of time for all SRF samples.

**Figure 4 polymers-13-01856-f004:**
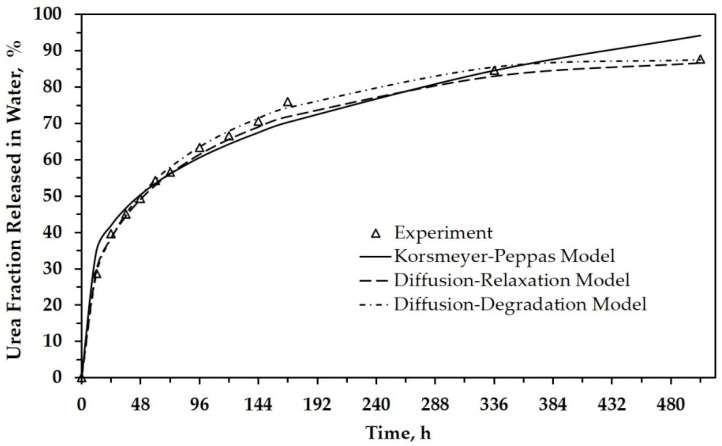
The urea fraction released in water as a function of time for SRF101, based on experimental and calculation data.

**Figure 5 polymers-13-01856-f005:**
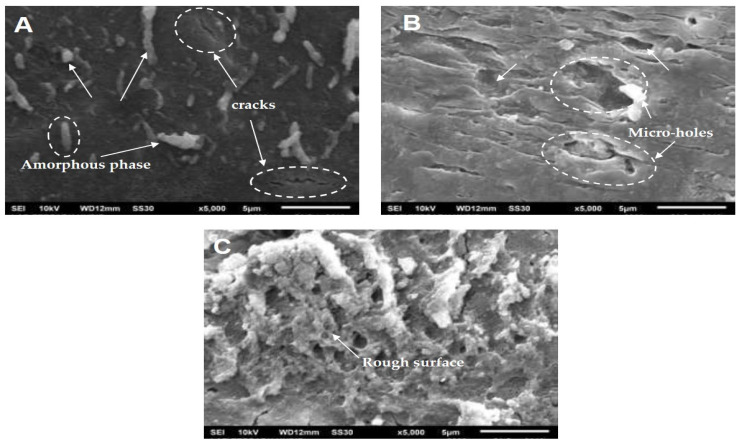
SEM images of neat PLA with different immersion times: 0 h (**A**), 168 h (**B**) and 504 h (**C**).

**Figure 6 polymers-13-01856-f006:**
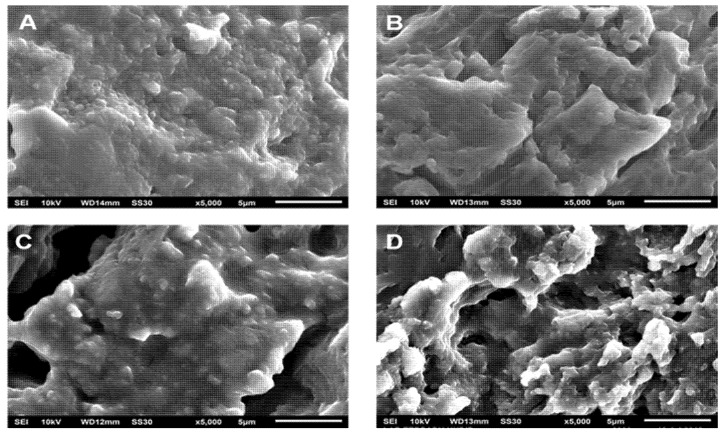
Morphological changes of SRF201 during immersion at different times: 0 h (**A**), 96 h (**B**), 168 h (**C**), 504 h (**D**).

**Table 1 polymers-13-01856-t001:** Nomenclature of samples.

Sample	Polymerization Time, h	Average *M*_W_ of PLA, Da	Urea Content in 3 g of SRF, g
Neat PLA	16	6015.2	0
SRF101	16	6015.2	0.01
SRF201	24	10,264.7	0.01
SRF301	32	13,564.2	0.01
SRF203	24	10,264.7	0.03
SRF205	24	10,264.7	0.05

**Table 2 polymers-13-01856-t002:** Changes in acidity (pH) during the urea release test.

Time, h	Acidity (pH) of Solution					
	Neat PLA	SRF101	SRF201	SRF301	SRF203	SRF205
0	6.80 ± 0.01	6.80 ± 0.01	6.80 ± 0.01	6.80 ± 0.01	6.80 ± 0.01	6.80 ± 0.01
12	6.79 ± 0.02	7.27 ± 0.05	6.95 ± 0.08	6.97 ± 0.05	7.19 ± 0.05	7.26 ± 0.06
24	6.78 ± 0.02	7.39 ± 0.05	7.08 ± 0.03	7.06 ± 0.05	7.33 ± 0.06	7.43 ± 0.04
32	6.76 ± 0.02	7.52 ± 0.06	7.15 ± 0.03	7.16 ± 0.06	7.41 ± 0.06	7.59 ± 0.03
48	6.75 ± 0.01	7.55 ± 0.03	7.20 ± 0.03	7.23 ± 0.07	7.54 ± 0.05	7.70 ± 0.03
60	6.74 ± 0.01	7.38 ± 0.03	7.29 ± 0.08	7.33 ± 0.05	7.67 ± 0.04	7.81 ± 0.03
72	6.72 ± 0.01	7.22 ± 0.05	7.35 ± 0.08	7.34 ± 0.03	7.71 ± 0.06	7.63 ± 0.05
96	6.69 ± 0.02	7.19 ± 0.04	7.33 ± 0.07	7.29 ± 0.04	7.57 ± 0.06	7.58 ± 0.05
120	6.62 ± 0.04	7.16 ± 0.03	7.25 ± 0.07	7.20 ± 0.06	7.51 ± 0.08	7.52 ± 0.03
144	6.50 ± 0.04	7.08 ± 0.05	7.16 ± 0.05	7.15 ± 0.08	7.50 ± 0.06	7.48 ± 0.04
168	6.45 ± 0.03	7.01 ± 0.03	7.10 ± 0.06	7.13 ± 0.05	7.44 ± 0.05	7.43 ± 0.02
336	6.34 ± 0.04	6.96 ± 0.05	7.04 ± 0.05	7.06 ± 0.04	7.38 ± 0.03	7.37 ± 0.03
504	6.20 ± 0.03	6.92 ± 0.04	6.99 ± 0.03	7.01 ± 0.09	7.31 ± 0.06	7.33 ± 0.06

**Table 3 polymers-13-01856-t003:** Diffusion parameters from the Korsmeyer–Peppas model.

Sample	Diffusion/Korsmeyer–Peppas Model			
	*k*	*n*	*R* ^2^	Type of Diffusion
Neat PLA	-	-	-	-
SRF101	0.1796 ± 0.0157	0.2663 ± 0.0173	0.9789	Fickian
SRF201	0.1406 ± 0.0155	0.3028 ± 0.0216	0.9721	Fickian
SRF301	0.1343 ± 0.0173	0.3043 ± 0.0251	0.9629	Fickian
SRF203	0.2142 ± 0.0262	0.2417 ± 0.0244	0.9553	Fickian
SRF205	0.2334 ± 0.0274	0.2315 ± 0.0235	0.9524	Fickian

**Table 4 polymers-13-01856-t004:** Diffusion and relaxation parameters from the diffusion–relaxation model.

Sample	Diffusion–Relaxation Model		
	*k* _1_	*k* _2_	*R* ^2^
Neat PLA	-	-	-
SRF101	0.1119 ± 1.98 × 10^−3^	−0.0036 ± 1.84 × 10^−4^	0.9949
SRF201	0.1015 ± 2.56 × 10^−3^	−0.0028 ± 2.39 × 10^−4^	0.9916
SRF301	0.0981 ± 3.16 × 10^−3^	−0.0028 ± 2.95 × 10^−4^	0.9865
SRF203	0.1284 ± 2.27 × 10^−3^	−0.0047 ± 2.11 × 10^−4^	0.9972
SRF205	0.1352 ± 1.95 × 10^−3^	−0.0051 ± 1.81 × 10^−4^	0.9969

**Table 5 polymers-13-01856-t005:** Diffusion and degradation parameters from the diffusion–degradation model.

Sample	Diffusion–Degradation Model				
	*a*	*b*	*c*	*d*	*R^2^*
Neat PLA	-	-	-	-	-
SRF101	0.0932 ± 2.67 × 10^−3^	−3.11 × 10^−3^ ± 3.86 × 10^−4^	2.32 × 10^−6^ ± 1.26 × 10^−7^	−1.80 × 10^−9^ ± 1.51 × 10^−7^	0.9987
SRF201	0.0705 ± 2.84 × 10^−3^	−7.54 × 10^−4^ ± 4.12 × 10^−5^	−3.33 × 10^−6^ ± 1.35 × 10^−7^	3.92 × 10^−9^ ± 1.61 × 10^−10^	0.9985
SRF301	0.0656 ± 5.77 × 10^−3^	−6.83 × 10^−5^ ± 8.36 × 10^−5^	−5.80 × 10^−6^ ± 2.74 × 10^−7^	6.91 × 10^−9^ ± 2.74 × 10^−10^	0.9936
SRF203	0.0986 ± 4.19 × 10^−3^	−2.91 × 10^−3^ ± 6.07 × 10^−4^	−6.21 × 10^−8^ ± 1.99 × 10^−9^	1.10 × 10^−9^ ± 2.37 × 10^−10^	0.9972
SRF205	0.1028 ± 3.10 × 10^−3^	−2.92 × 10^−3^ ± 4.48 × 10^−4^	−1.23 × 10^−6^ ± 1.47 × 10^−7^	2.92 × 10^−9^ ± 1.75 × 10^−10^	0.9984

## Data Availability

Data are contained within the article.

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
