# Peer review of "A Slow-Release Fertilizer of Urea Prepared via Melt Blending with Degradable Poly(lactic acid): Formulation and Release Mechanisms"

_polymers, 2021, doi:10.3390/polym13111856_

Round 1

Reviewer 1 Report

Authors present results from release experiments on poly(lactic acid) – urea blends with the aim to analyse potential application as environmental-friendly fertilizer.

The present manuscript is severely plagued by problems in the presentation and readability. There exist numerous grammar and syntax error which make some parts of the manuscript very difficult to understand. The manuscript thus requires major revisions before being considered for publication in Polymers or any journal in the field of polymer science. Furthermore, many of the presented trends are rather expected and the interpretation of data is quite lacking.

) As mentioned above the manuscript requires a very careful check and editing from preferably a native speaker and one familiar with the terminology used. “Simulation” used in some instances should be removed and be replaced by “data analysis”.

) Presentation problems exist in various parts. For example, concentration data are presented in Table 2 in a format which is almost impossible to understand. Then, corresponding urea fractions appear as Fig. 3; out of this set one curve is fitted with one model in Fig. 4 then the same curve with all models in Fig. 7. Accordingly, there is no point in presenting Fig. 4 with just one model, neither the tables with this format that makes it impossible to read.

) To improve the reading flow of the manuscript and to avoid repeating the same data in different figures all three models could be introduced as early as in Materials and Methods with explanation of the corresponding parameters. Then, they could be used to fit the experimental data showing the fittings on selected systems for all models.

) The need for the separation of regions A and B in Fig. 3 is far from clear. Such a trend of decaying the percentage released is not expected as time increases?

) SEM images are presented but conceptually they appear disconnected with the rest of the manuscript and with the trends on the urea released.

) It is not clear how the value for m is decided in Eq. 2. If the authors use it as a fitting parameter in the model what would be the corresponding value compared to the current one of 0.4625 ?

) That the diffusion is the dominant factor is not an expected finding for the diffusion-relaxation and diffusion-degradation models? Also given the very low values of corresponding fitting parameters the better match of these two models seems to be a matter of just having a larger set of parameters in the formula.

) “Novel” could be removed from the title.  

Reviewer 2 Report

In this work, poly(lactic acid)s with different low molecular weights were synthesized by direct polycondensation, and then melt compounded with urea to formulate slow-release fertilizer (SRF). The influence of molecular weight of PLA as a matrix and urea composition in SRF towards release kinetics in water at 30 oC were studied in detail. The overall framework of this work is clear, and the research content of each part is relatively complete. However, there are some big problems for this research, including insufficient innovation, some key performances needed to be further investigated and other problems. If the relevant performance data is supplemented on the basis of current work, this article will be more perfect.

  1. As described in the article, the degradation mechanism of PLA has been clearly reported in the literature. The biggest innovation of this work should be to make good use of the degradation mechanism of PLA to guide the design and preparation of slow-release fertilizer (SRF), and adjust the slow-release rate according to external factors. However, the preparation method of present work is randomly melt-blending of PLA and urea, without special PLA coating or other morphological designs, which makes slow-release regulation and mechanism research less meaningful. Because urea itself will be released directly even if it is blended with PLA, it is difficult to know the slow-release effect in the influencing factors analysis. For example, in the No.337 line, researchers thought that utilizing the higher Mw of PLA can inhibit the urea release. There are two issues to be considered here: one question is what is the effect mechanism of molecular weight on the slow-release? Another question is whether the mixed coating form of PLA/urea, urea content in the blended mixture will have a more significant impact on slow-release? In fact, in Figure 3, it can be seen that the effect of PLA molecular weight is not obvious, while the effect of urea addition is greater. To a certain extent, this shows that the PLA slow-release effect has not been fully utilized.
  2. Molecular weight of PLA is an important performance index for this work. This value was determined using the Mark-Houwink-Sakurada equation in this manuscript. This method is an indirect method of characterizing molecular weight of polymer, and the viscosity-average molecular weight is given. It cannot accurately reflect the weight average molecular weight of PLA. Thus, it is recommended to supplement the Mw values characterized by GPC method or light scattering method. In addition, please note the accurate molecular weight identification, the letter w is subscript in Mw.
  3. Researchers have always emphasized this fertilizer usage does not leave residues that damage the soil structure and nutrient balance in the soil. But the present work only gives the static release test in water at 30 oC, but lacks the results of experiments burred in soil.
